# Comparison of 6.0 mm versus 6.5 mm Optical Zone on Visual Outcomes after LASIK

**DOI:** 10.3390/jcm10173776

**Published:** 2021-08-24

**Authors:** Majid Moshirfar, Rachel Huynh, Nour Bundogji, Alyson N. Tukan, Thomas M. Sant, Shannon E. McCabe, William B. West, Kirk Drennan, Yasmyne C. Ronquillo, Phillip C. Hoopes

**Affiliations:** 1Hoopes Vision Research Center, Hoopes Vision, Draper, UT 84020, USA; shannonemccabe@gmail.com (S.E.M.); kdrennan@hoopesvision.com (K.D.); yronquillo@hoopesvision.com (Y.C.R.); pch@hoopesvision.com (P.C.H.); 2John A. Moran Eye Center, Department of Ophthalmology and Visual Sciences, University of Utah, Salt Lake City, UT 84132, USA; 3Utah Lions Eye Bank, Murray, UT 84107, USA; 4University of Utah School of Medicine, Salt Lake City, UT 84132, USA; Rachel.huynh@hsc.utah.edu (R.H.); Thomas.Sant@hsc.utah.edu (T.M.S.); wbwest@gwu.edu (W.B.W.J.); 5University of Arizona College of Medicine Phoenix, Phoenix, AZ 85004, USA; nbundogji@gmail.com (N.B.); atukan@email.arizona.edu (A.N.T.); 6Mission Hills Eye Center, Pleasant Hill, CA 94523, USA

**Keywords:** optical zone size, 6.0 mm, 6.5 mm, LASIK, Allegretto, refractive outcomes, myopia, astigmatism, dry eye syndrome, SPK

## Abstract

Previous studies have demonstrated safety and efficacy using 6.0 and 6.5 mm optical zones in the WaveLight EX500 Excimer Laser System but have not evaluated if differing optical zone sizes influence refractive outcomes. This study examines visual outcomes between two study populations undergoing LASIK with either a 6.0 mm (1332 patients) or 6.5 mm (1332 patients) optical zone. Outcomes were further stratified by severity of myopia (low, moderate, and high) and astigmatism (low and high). Patients were matched by age and preoperative manifest sphere and cylinder. Postoperative measurements were then compared. The 6.5 mm group demonstrated better postoperative manifest refractive spherical equivalent (MRSE), manifest sphere, and absolute value of the difference in actual and target spherical equivalent refraction (|∆ SEQ|), within the total population, moderate myopia, and low astigmatism groups, but this did not lead to improved postoperative uncorrected distance visual acuity (UDVA) or best corrected distance visual acuity (CDVA). Though astigmatic correction and postoperative angle of error were similar between optical zone sizes, they were significantly worse with high myopia. Overall, this study demonstrates differences in visual outcomes between the 6.0 and 6.5 mm optical zone sizes that may warrant consideration; however, essentially, the results are comparable between them.

## 1. Introduction

Excimer laser surgery was first experimented on cadaveric eyes in 1983 [1], patented for refractive use in live corneas in 1989 [2], and used with laser in situ keratomileusis (LASIK) in 1990 [3]. Excimer lasers have continued to evolve, transitioning from full beam to scanning slit and flying spot beam delivery for improved precision, along with increased laser ablation frequencies to shorten treatment time [4]. Additionally, blend zones and larger optical zones have improved visual outcomes in refractive surgery [5,6]. Historically, excimer lasers used a 4–5 mm optical zone. However, post-LASIK epithelial and stromal remodeling at the transition between ablated and untreated cornea created corneal irregularities, reducing the effective optical zone and refractive outcomes while increasing the risk of myopic regression [6,7,8,9]. Implementation of blend zones created smooth transition points, decreasing corneal remodeling and aberrations [5,10], and larger optical zones reduced the chance of early myopic regression [11], hyperopic shift [12], and postoperative higher-order aberrations (HOAs) [13,14]. As a result, subjective symptoms like decreased contrast sensitivity and glare have been reduced [15]. Thus, the use of larger optical zones (commonly 6.0 and 6.5 mm) along with blend zones have become standard practice, creating a total treatment zone around 8.5 to 9 mm in diameter.

The WaveLight Allegretto Wave Excimer Laser System (Alcon, Fort Worth, TX, USA), a 200 Hertz (Hz) laser, was approved by the Food and Drug Administration (FDA) in October 2003. Safety and efficacy trials were conducted using a 6.5 mm optical zone [16], though the device was approved for 6.0 and 6.5 mm. Similarly, the Allegretto Wave Eye-Q Excimer Laser System (Alcon, Fort Worth, TX, USA), approved in June 2006, demonstrated the safety of 400 Hz using a 6.5 mm optical zone [17,18]. In 2011, the WaveLight EX500 Excimer Laser System (WL EX500) (Alcon, Fort Worth, TX, USA) was approved for use in 6.0 to 6.5 mm optical zones [18]. Safety trials using a 500 Hz pulse rate were conducted on porcine and donor human eyes [4,19,20], and a small clinical trial showed no adverse outcomes [21]. However, the clinical use of the WL EX500 on 6.0 and 6.5 mm optical zones is based on clinical trials from the previously approved lasers. While these lasers were determined to be safe for both optical zone sizes, the studies did not examine whether outcomes differed between them. 

The present study is a large-scale evaluation of visual outcomes in post-LASIK eyes to compare the predictability, efficacy, and safety of the WL EX500 using 6.0 and 6.5 mm optical zones with a 1.25 mm blend zone. We further evaluate whether preoperative severity of myopia or astigmatism contributes to the success of using a 6.0 or 6.5 mm optical zone. An additional analysis was performed on patient-reported glare/halos, dry eye, and post-procedural satisfaction, as well as the presence of superficial punctate keratitis (SPK) on a slit lamp exam. These additional measures were used to compare the different optical zone sizes’ subjective and objective success rates. 

## 2. Materials and Methods

We retrospectively analyzed de-identified data in a chart review of protected medical records from a tertiary refractive surgery center (Hoopes Vision, Draper, UT, USA). Myopic patients underwent LASIK on the WL EX500 platform with the iFS^®^ Advanced Femtosecond Laser (Johnson and Johnson Vision, Santa Ana, CA, USA) by five surgeons from April 2014 to March 2019 were included. Inclusion criteria were: Aged 18 years and older and best corrected distance visual acuity (CDVA) of 20/20 or better preoperatively. Exclusion criteria included: Bioptics patients, severe preoperative myopia (manifest sphere ≥9.0 diopters [D]), severe preoperative astigmatism (cylinder ≥ 4.0 D), or less than twelve months of postoperative follow-up. Patients in this study did not undergo any enhancements until after the first twelve-month postoperative period for better comparison of the efficacy of the two optical zone sizes. 

Standard preoperative LASIK evaluations were performed. For this study, we analyzed the following: preoperative manifest refraction (MR), manifest refraction spherical equivalent (MRSE), uncorrected distance visual acuity (UDVA), CDVA, the absolute value of the angle of error (|Angle of error|), the absolute value of the difference in actual and target spherical equivalent refraction (|∆ SEQ|), and presence of SPK. Standard postoperative objective outcomes measured at each postoperative visit were: (1) UDVA; (2) CDVA; (3) MR; (4) SPK. The Efficacy Index ($\frac{UDVA\;postoperative}{CDVA\;preoperative}$) and Safety Index ($\frac{CDVA\;postoperative}{CDVA\;preoperative}$) were calculated.

For the surgical procedure, iFS IntraLase was used to create flaps, with an average flap thickness of 105 μm with a 9 mm flap diameter using the following laser settings of 150 kHz, 1.15 μJ bed energy, 2.00 μJ side cut energy, and pocket enabled. This was followed by treatment with WL EX500 laser for all patients included in this study. A 6.0 or 6.5 mm optical zone was chosen based on random selection with a blend zone of 1.25 mm. The attempted correction for the treatment of myopic astigmatism was based on manifest refraction using the nomogram recommendation from the manufacturer. Postoperatively, patients were treated with a standard regimen of ofloxacin 0.3% (Allergan Inc., Irvine, CA, USA) and prednisolone acetate 1% (Allergan Inc., Irvine, CA, USA). Postoperative visits were conducted one day, one week, one month, three months, six months, and twelve months following the operation date. 

### 2.1. Patient Stratification and Matching

Patients were stratified based on optical zone size (6.0 and 6.5 mm), level of myopia, and magnitude of cylinder. Myopia was categorized using the following criteria: Low = 0 to <−3.0 D; moderate = −3.0 to <−6.0 D; and high = −6.0 to <−9.0 D. Absolute astigmatism was categorized using the following criteria: Low = <−2 D; high = −2 to <−4 D. One eye was randomly chosen from each subject for data analysis. Patients in the 6.0 and 6.5 mm optical zone groups were matched for preoperative sphere, cylinder, and age.

### 2.2. Vector Analysis

Vector analysis was performed on preoperative and twelve-month postoperative manifest refractions by comparing the predicted refraction at the corneal plane. Predicted postoperative refraction was selected as plano with zero cylinder in the corresponding preoperative axis of the preoperative manifest cylinder unless targeted monovision was specified. The American Society of Cataract and Refractive Surgery astigmatism double-angle plot tool was used to perform a vector analysis on data, which were compiled and presented using the methods described by Abulafia et al. [22].

### 2.3. Patient Reported Outcomes and Qualitative Data Collection

A subset of eyes from the 6.0 mm (*n* = 250 patients) and 6.5 mm (*n* = 250 patients) optical zone groups were retrospectively assessed for subjective symptoms of dry eyes, glare/halos, and overall satisfaction following LASIK documented at any point preoperatively and within the 1-year follow-up period within their charts. They were also evaluated for level of SPK on slit-lamp as well as postoperative UDVA and CDVA. For this patient subset, 250 patients receiving 6.0 and 6.5 mm optical zone sizes were selected prior to de-identification, with only one eye per patient included in the de-identified analysis. These patients had the data needed for analysis. Postoperative symptoms were reported relative to preoperative symptoms.

### 2.4. Statistical Analysis

Patient matching for sphere, cylinder, and age was performed using the Exact Matching and Nearest Neighbor Matching techniques of Propensity Score Matching in RStudio (RStudio, Inc. Released 2018. RStudio for Macintosh, Version 1.1.456. Boston, MA, USA: RStudio Inc.). Summary descriptive statistics were calculated for all variables. For continuous variables measured, such as the preoperative and postoperative visual outcomes, a two-tailed independent samples *t*-test was used to compare the 6.0 and 6.5 mm groups. Additionally, a modified T-squared distribution was used in the vector analysis.

All discrete variables in the total population study, such as preoperative baseline characteristics or postoperative line changes in UDVA and CDVA, were analyzed with a Pearson’s Chi-square. A post-hoc Bonferroni adjustment was further used to determine statistical significance. A Pearson’s Chi-square or Fisher’s exact test with post-hoc Hochberg method was used on discrete variables obtained from the qualitative study (postoperative satisfaction, need for enhancement, glare/halos, dry eye symptoms, and SPK). 

To evaluate differences between the proportion of each group achieving specific outcomes (e.g., ≤0.50 D residual cylinder), a test of given proportions was performed. Hotelling’s T-squared distribution was used to determine statistical significance of the vector analyses. To compare the differences in linear regression of target induced astigmatism (TIA) vs. surgically induced astigmatism (SIA) in the subgroups, a pairwise comparison of the slopes of fitted lines was used. For all statistical analyses, *p*-values of <0.05 were determined to be statistically significant.

### 2.5. Ethics Approval and Informed Consent

All patients were fully informed and consented to treatment. All methods and procedures followed the tenets of the Declaration of Helsinki and were approved by the Hoopes Vision Ethics Review Board. The Biomedical Research Alliance of New York (Brany) Institutional Review Board (New York) approved this study using de-identified data.

## 3. Results

### 3.1. Preoperative Characteristics

After stratification and matching, there were 1332 patients within each group with no significant differences in age, sex, MRSE, manifest sphere, manifest cylinder, and target refraction (Table 1).

### 3.2. Postoperative Outcomes for Total Population

#### 3.2.1. Visual Outcomes

Table 2 displays postoperative visual outcomes between the 6.0 and 6.5 mm groups. The MRSE was significantly more myopic in the 6.0 mm group (−0.16 ± 0.37 D) compared to the 6.5 mm group (−0.12 ± 0.35 D) (*p* = 0.01). Similarly, the manifest sphere was more myopic in the 6.0 mm group (6.0 mm = −0.03 ± 0.36 D; 6.5 mm = −0.01 ± 0.35 D; *p* = 0.01). The mean absolute difference in SEQ was greater in the 6.0 mm group (6.0 mm = 0.22 ± 0.23 D; 6.5 mm = 0.19 ± 0.20 D; *p* < 0.001). 

#### 3.2.2. UDVA and Efficacy Index

At twelve months postoperatively, 1072 eyes (87%) of the 6.0 mm group and 1085 eyes (88%) of the 6.5 mm group achieved UDVA of 20/20 or better (Figure 1A). Additionally, the Efficacy Index was 1.04 and 1.02 for the 6.0 and 6.5 mm groups, respectively.

#### 3.2.3. CDVA and Safety Index

The UDVA was the same or better than the preoperative CDVA in 87% of eyes in the 6.0 mm group and 89% of eyes in the 6.5 mm group at twelve months (Figure 1B). One line of CDVA was gained postoperatively in 50% of eyes in the 6.0 mm group and 49% of eyes in the 6.5 mm group (Figure 1C). Within the 6.0 and 6.5 mm groups, 2% and 4% of eyes, respectively, lost one line of CDVA (Figure 1C), most of which went from 20/15 preoperatively to 20/20 postoperatively. The lack of further refraction once a patient reached a 20/20 CDVA in follow up visits likely accounts for a majority of the loss of one line of CDVA within the 6.0 and 6.5 mm groups. The Safety Index for the 6.0 mm group was 1.14, while the 6.5 mm group had a Safety Index of 1.13 over the same postoperative timeframe.

#### 3.2.4. Stability and Predictability

The 6.0 and 6.5 mm groups showed predictable visual outcomes, demonstrated by UDVA at twelve months (Figure 1D). The slopes of attempted versus achieved MRSE were 0.99 and 1.01 for 6.0 and 6.5 mm groups, respectively. At twelve months, reaching within 0.5 D of target MRSE was statistically different between 6.0 mm (93%) and 6.5 mm (97%) (*p* < 0.001). No significant difference was observed between the rate of reaching within 1.00 D of target MRSE in the 6.0 mm (99%) and 6.5 mm group (100%) (*p* = 0.24) (Figure 1E). Stability is demonstrated in Figure 1F, with 9% of eyes experiencing a change in MRSE > 0.5 D in the 6.0 mm group and 10% in the 6.5 mm group. The MRSE was significantly worse in the 6.0 group across all time points, except at six months (1 mo *p* = 0.013; 3 mo *p* = 0.019; 6 mo *p* = 0.053; 12 mo *p* = 0.005). Further, the one-month to the twelve-month MRSE decreased by −0.16 D in the 6.0 mm group and −0.12 D in the 6.5 mm group.

Vector analysis of the 6.0 and 6.5 mm groups at 12 months can be seen in Figure 2. Figure 2A shows the preoperative and postoperative refractive astigmatism results. The preoperative refractive astigmatism centroid was 0.16 ± 1.02 D at 89° and the twelve-month postoperative centroid was 0.10 ± 0.37 D at 92° for the 6.0 mm group (Figure 2B). The preoperative refractive astigmatism centroid for the 6.5 mm group was 0.35 ± 0.96 D at 89° and the twelve-month postoperative centroid was 0.13 ± 0.35 D at 89° (Figure 2C). Postoperative refractive astigmatic prediction errors in the corneal plane are plotted in Figure 2D,E. At twelve months, the prediction error was ≤1.00 D in 99% of eyes and ≤0.50 D in 90% of eyes for both groups (Figure 1G). The TIA and SIA at twelve months are shown in Figure 1H. The mean angle of error was 1.1 ± 18.5° for 6.0 mm and 0.5 ± 17.2° for 6.5 mm (Figure 1I), indicating good postoperative astigmatic outcomes.

### 3.3. Outcomes Based on Level of Myopia

The UDVA and CDVA in the 6.0 and 6.5 mm groups did not significantly differ when eyes were stratified by preoperative myopia (Table 3). Of eyes categorized as low myopia, |∆ SEQ| was significantly greater in the 6.0 mm group (0.20 ± 0.23 D) compared to the 6.5 mm group (0.17 ± 0.18 D) (*p* = 0.01). Of eyes categorized as moderate myopia, the 6.0 mm group was significantly more myopic postoperatively (MRSE −0.18 ± 0.39 D; sphere −0.04 ± 0.38 D) compared to the 6.5 mm group (MRSE −0.14 ± 0.33 D; sphere −0.03 ± 0.33 D) (MRSE *p* = 0.03; sphere *p* = 0.02). The change in mean absolute SEQ was significantly greater in the 6.0 mm group (0.23 ± 0.22 D) compared to the 6.5 mm group (0.18 ± 0.20 D) (*p* < 0.001). There were no statistically significant differences in visual outcomes in eyes categorized by high myopia. A visual comparison of outcomes in using a 6.0 and 6.5 mm optical zone can be seen for the low myopia group in Figure A1, the moderate myopia group in Figure A2, and the high myopia group in Figure A3.

At twelve months postoperatively, 87% of the 6.0 mm group and 88.0% of the 6.5 mm group achieved UDVA of 20/20 or better (Figure A1A). However, the 6.0 mm group had a higher percentage (46%) of patients with a “gain of one line” when comparing their postoperative UDVA to preoperative CDVA than the 6.5 mm group (40%), though this did not achieve statistical significance (*p* = 0.05) (Figure A1B). When comparing postoperative CDVA to preoperative CDVA, 53% of eyes in the 6.0 mm group and 48% of eyes in the 6.5 mm group gained one line (Figure A1C).

More eyes gained one line in postoperative UDVA and postoperative CDVA in the 6.5 mm group compared to the 6.0 mm group (Figure A2 and Figure A3), though none reached statistical significance. Comparison of the distributions in postoperative UDVA and CDVA in the total population and levels of myopia and astigmatism only showed significance in low myopia (*p* = 0.002), with a post-hoc analysis attributing the significance to the “loss of three or more lines” category. 

Regarding refractive astigmatism, regardless of optical zone size, fewer postoperative patients reached ≤0.50 D of residual manifest cylinder as the level of myopia increased (*p* < 0.001 for both 6.0 and 6.5 mm) (Figure A1G, Figure A2G and Figure A3G). Furthermore, there was decreased efficacy in astigmatic correction in the high myopia group compared to low and moderate myopia groups (for 6.0 mm = *p* < 0.001 comparing high to low, *p* = 0.001 comparing high to moderate; for 6.5 mm = *p* = 0.0001 comparing high to low, *p* = 0.002 comparing high to moderate) (Figure A1H, Figure A2H and Figure A3H). As the severity of myopia increased, there was also an increase in rates of the angle of error ≥±15° for refractive astigmatism (*p* < 0.001 for both 6.0 and 6.5 mm) (Figure A1I, Figure A2I, and Figure A3I).

### 3.4. Outcomes Based on Preoperative Level of Astigmatism

The UDVA and CDVA in the 6.0 and 6.5 mm groups did not significantly differ when eyes were stratified by preoperative astigmatism (Table 4). Within eyes categorized as low astigmatism, the 6.0 mm group was significantly more myopic postoperatively (MRSE −0.16 ± 0.38 D; sphere −0.04 ± 0.36 D) compared to the 6.5 mm group (MRSE −0.12 ± 0.35 D; sphere −0.01 ± 0.34 D) (MRSE *p* = 0.002; sphere *p* = 0.002). The|∆ SEQ| at twelve-months was significantly greater in the 6.0 mm group compared to the 6.5 mm group for both low astigmatism (6.0 mm = 0.21 ± 0.23 D; 6.5 mm = 0.18 ± 0.19 D; *p* < 0.001) and high astigmatism (6.0 mm = 0.28 ± 0.26 D; 6.5 mm = 0.20 ± 0.20 D; *p* = 0.01). A visual comparison of the outcomes in using a 6.0 and 6.5 mm optical zone can be seen for the low astigmatism group in Figure A4 and the high astigmatism group in Figure A5.

For the low astigmatism category, more eyes were within ≤0.50 D of targeted MRSE in the 6.5 mm group compared to the 6.0 mm group (*p* = 0.008) (Figure A4E), though there was no difference found in the high astigmatism category (*p* = 0.16) (Figure A5E). Figure 3 and Figure 4 provide a visual comparison between low and high astigmatism of the preoperative and postoperative centroids for 6.0 or 6.5 mm, respectively. Vector analysis comparing postoperative centroids showed better postoperative results in the low astigmatism group (6.0 and 6.5 mm = *p* < 0.001) (Figure 3A and Figure 4A). In both 6.0 and 6.5 mm groups, those that started with low astigmatism had a significantly higher likelihood of reaching ≤0.50 D of residual cylinder compared to the high astigmatism group (6.0 and 6.5 mm = *p* < 0.001) (Figure A4G and Figure A5G). However, astigmatism was significantly under corrected in the low astigmatism category compared to the high astigmatism category in the 6.0 mm group (*p* = 0.007), though not in the 6.5 mm group (*p* = 0.19) (Figure A4H and Figure A5H). Regarding angle of error, the low astigmatism category was statistically more likely to have an angle of error ≥ ±15° than the high astigmatism category (6.0 and 6.5 mm = *p* < 0.001) (Figure A4I and Figure A5I).

### 3.5. Patient-Reported Outcomes and Qualitative Data Collection

Patients reported higher postoperative satisfaction, rated as “good,” in the 6.5 mm group (98%) compared to the 6.0 mm group (93.2%) (*p* = 0.03) (Figure 5A). In the 6.0 mm group, 6.4% of patients had “okay” satisfaction versus 2% in the 6.5 mm group. Only 0.4% of patients were “unhappy” with their visual outcome in the 6.0 mm group, while none were “unhappy” in the 6.5 mm group. Eyes in the 6.0 mm group (4.8%) were more likely to undergo LASIK enhancement than in the 6.5 mm group (1.2%) (*p* = 0.04) (Figure 5B). However, despite decreased satisfaction and increased rate of enhancement in the 6.0 mm group, patients did not have significantly more frequent experiences of postoperative glare/halos (*p* = 0.23) or dry eyes (*p* = 0.54) (Figure 5C,E). They also did not have an increased severity of postoperative glare/halos (*p* = 0.18) or dry eyes (*p* = 0.88) (Figure 5D,F). Objective measures of dryness, recorded as the presence of SPK, also showed no significant change between preoperative and postoperative visits in both groups with frequency (*p* = 0.87) or severity (*p* = 0.61) of SPK (Figure 5G). There was also no significant difference in postoperative UCVA (*p* = 0.08) or CDVA (*p* = 0.45) (Figure 5I,J).

## 4. Discussion

This study assessed visual outcomes between patients undergoing LASIK with a 6.0 or 6.5 mm optical zone using the WL EX500. While the WL EX500 is approved for use with a 6.0 or 6.5 mm optical zone, the initial clinical trials showing appropriate safety and efficacy only used a 6.5 mm optical zone in the 200 Hz WaveLight Allegretto Wave Excimer Laser. Recent studies have shown that the WL EX500 achieves excellent visual outcomes, even for patients with high myopia [23], and is safe, effective, and predictable for the correction of refractive error [24]. However, these investigations failed to examine the effects of different optical zones on safety, efficacy, predictability, and stability.

Our study benefits from a large sample size, with 1332 patients in each group with random selection of one eye per subject. Matched samples to manifest sphere, manifest cylinder, and age also allowed for control of any discrepancies resulting from a different distribution of myopia or astigmatism within each group. Further, none of the eyes in this study population underwent enhancements during the first twelve-month postoperative period in this study.

The results show that the size of optical zones influenced the rates of reaching emmetropia in the total study population, as demonstrated by the statistically significant difference in MRSE (at 1, 3, and 12 months), manifest sphere, and|∆ SEQ|, with a 6.5 mm optical zone having more favorable results. Furthermore, when stratified by level of myopia or astigmatism, MRSE, manifest sphere, and|∆ SEQ| were also significantly better with a 6.5 mm optical zone in the moderate myopia and low astigmatism subsets. These differences could potentially be attributed to a reduction in spherical aberrations and/or other HOAs [13,14]. 

Additionally, the two optical zone sizes were not statistically different in terms of the logMAR of UDVA and CDVA at 12 months. In one study, Ozulken similarly found no difference in UDVA and CDVA when evaluating 6.5 and 7.0 mm optical zones in photorefractive keratectomy [25]. However, our low myopia group did show a significant difference with the categorization of postoperative UDVA compared to preoperative CDVA. Within the 6.0 mm group, there was more under correction in UDVA compared to the 6.5 mm group. This is based on the finding that while 2% of the 6.0 mm group had a postoperative UDVA of ≤20/50, no patients in the 6.5 mm group were ≤20/50, potentially indicating lower rates of reaching desired UDVA in the 6.0 mm group. 

Despite larger optical zones improving visual acuity and reducing HOAs, larger optical zones require more tissue removal to achieve the same refractive power, leading to a smaller residual stromal bed [26] and limited potential for future enhancements. For every attempted diopter of myopic correction, an additional 3−4 microns of tissue is ablated with the 6.5 mm optic zone compared to the 6.0 mm optic zone, which may potentially increase the risk of ectasia [27] and forward displacement of the posterior cornea [28]. On the other hand, our data reflected a significant difference in rates of enhancement one year after LASIK, with the 6.0 mm group having higher rates of enhancement. Furthermore, the level of SEQ, residual manifest sphere, and MRSE were smaller and closer to emmetropia for the 6.5 mm group, as 4.8% of patients in the 6.0 mm group underwent enhancements compared to the 1.2% in the 6.5 mm group. While a 6.5 mm optical zone leaves less tissue for enhancement, our data suggest it also has lower rates of requiring postoperative enhancement. 

Increased laser treatment depth has been associated with a higher risk of developing dry eye symptoms [29]. The presence of dry eye symptoms in LASIK is also associated with a higher incidence of SPK [30]. This study found that the rates of postoperative dry eye symptoms were not statistically different between optical zone sizes, with 22% in the 6.0 mm group and 26% patients in the 6.5 mm group. Further, only 1.2% of the 6.0 mm group and 0.4% of the 6.5 mm group rated the severity of their dry eyes as “moderate” and no patients in either group rated it as “severe.” This is further reflected in the low rates of postoperative SPK, where 4.8% in both 6.0 and 6.5 mm groups had “mild” SPK. Therefore, our results do not indicate that larger optical zones increase the rate of postoperative dry eye.

Though this study benefits from a large sample size, the retrospective nature of the study limited the postoperative outcomes that we could evaluate. Specifically, quantitative measurements of HOAs, contrast sensitivity, spherical aberrations, keratometric changes, or pachymetry over time were not reported. Thus, the effect of optical zone size on the above parameters could not be fully investigated. Because these parameters have been shown in literature to be affected by optical zone size [13,14], a comparison of these outcomes between 6.0 and 6.5 mm optical zone sizes in a future study would be beneficial. Given the limitations of our reported data, we instead examined the subjective patient−reported outcomes (glare/halo, dry eyes, satisfaction levels) and SPK severity from slit lamp examination on a large cohort of patients (250 patients within each group). In the future, a prospective study or randomized control trial collecting the above parameters on a similarly large cohort would allow for better comparison of outcomes based on optical zone size. Ideally, a future randomized control trial with a double-blind placement of patients into each optical zone size cohort would be conducted to eliminate any selection bias and influence from history of dry eyes, HOAs, or other factors that could potentially influence a clinician’s decision regarding optical zone size.

To our knowledge, this is the first study to determine whether optical zone sizes demonstrate any pattern of differences when using the WL EX500. From a clinical standpoint, many factors are considered when attempting to choose the best approach to refractive surgery for patients. For example, clinicians may decide on using a 6.5 mm optical zone due to the slight improvement in the rate of achieving emmetropia, the higher rates of satisfaction, and potentially lower rates of enhancement. On the other hand, clinicians and patients may decide the slight benefit in outcomes in the 6.5 mm group is not worth the risks that come with removing more tissue. Our study also showed a significant difference between high myopia compared to low and moderate myopia in the accuracy of astigmatism correction, with more under correction of the cylinder seen in high myopia. Under correction of astigmatism was also seen in the low astigmatism category compared to the high astigmatism category, though only within the 6.0 mm group. Finally, fewer patients reached ≤0.50 D of residual cylinder with increasing myopia and astigmatism. These data may help guide conversations with patients on what can be expected of their postoperative correction depending on their preoperative sphere and cylinder.

## 5. Conclusions

In conclusion, these data demonstrate good efficacy and safety between both 6.0 and 6.5 mm optical zone sizes with the WL EX500. While some statistically significant differences exist between the two groups, ultimately, the outcomes with both optical zone sizes indicate that most patients achieve good postoperative results.

## Figures and Tables

**Figure 1 jcm-10-03776-f001:**
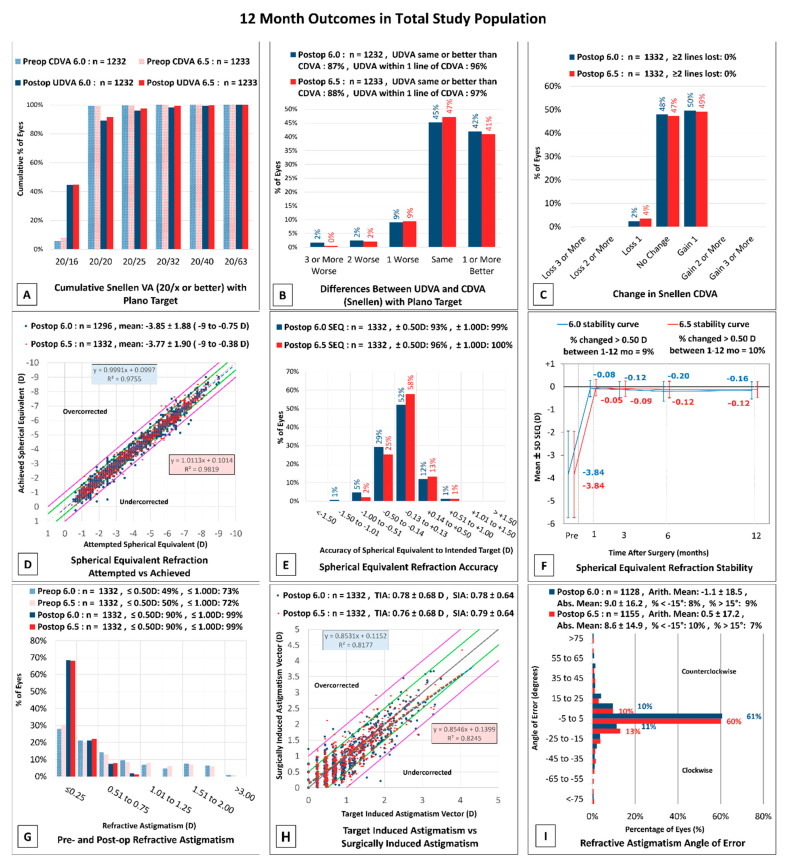
Standard reporting of visual and refractive outcomes for eyes at twelve months postoperatively between 6.0 and 6.5 mm groups. (**A**) Cumulative preoperative CDVA and postoperative UDVA; (**B**) difference between preoperative CDVA and postoperative UDVA; (**C**) change in Snellen CDVA; (**D**) attempted vs. achieved MRSE; (**E**) accuracy of MRSE; (**F**) MRSE stability at one, three, six, and 12 months; (**G**) comparison of preoperative and postoperative manifest cylinder; (**H**) target-induced vs. surgically-induced astigmatism; (**I**) angle of error.

**Figure 2 jcm-10-03776-f002:**
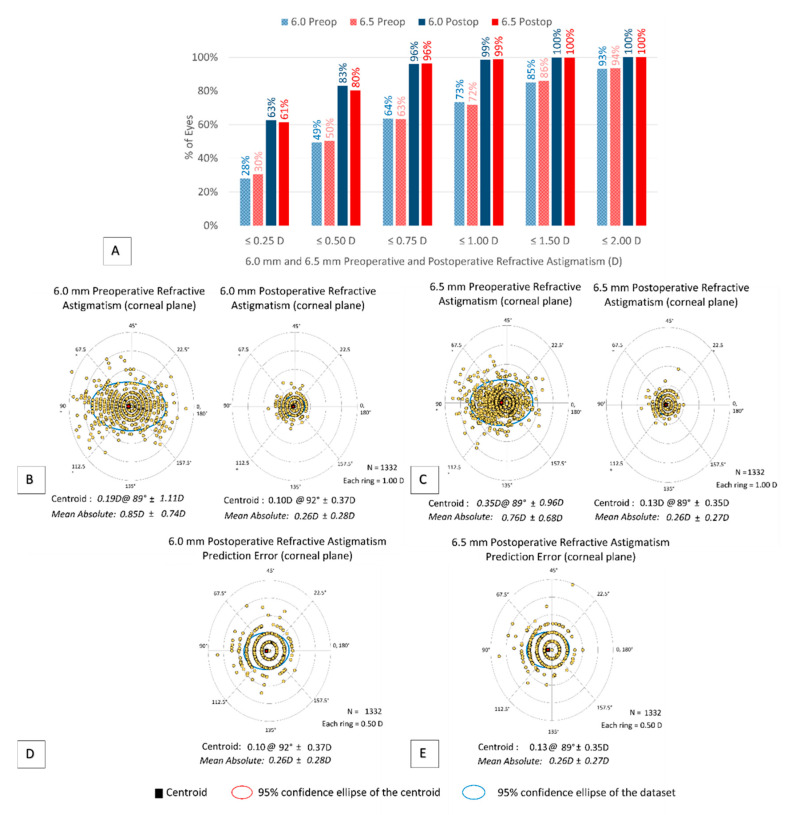
Vector analysis of 6.0 and 6.5 mm groups at 12 months. (**A**) Comparison of preoperative and postoperative manifest cylinder in diopters for 6.0 and 6.5 mm groups; (**B**) comparison of preoperative and postoperative refractive astigmatism for 6.0 mm group; (**C**) comparison of preoperative and postoperative refractive astigmatism for 6.5 mm group; (**D**) postoperative refractive astigmatism prediction error for 6.0 mm group; (**E**) postoperative refractive astigmatism prediction error for 6.5 mm group.

**Figure 3 jcm-10-03776-f003:**
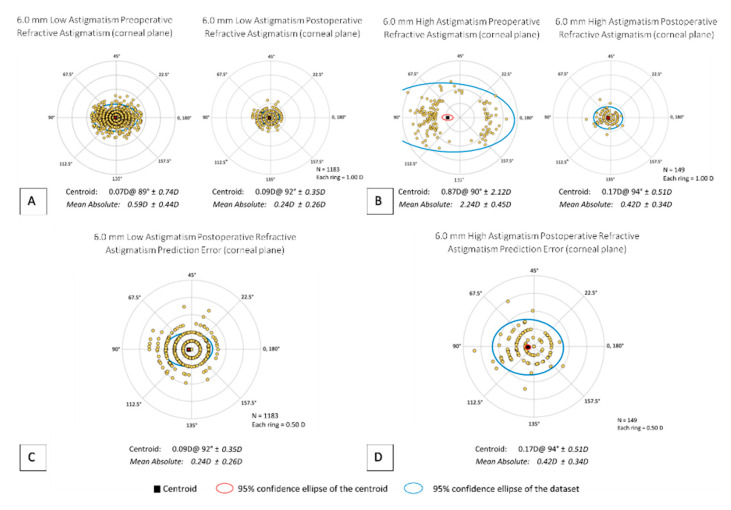
Vector analysis of 6.0 mm Low and High Astigmatism Subgroups. (**A**) Comparison of preoperative and postoperative refractive astigmatism for 6.0 mm Low Astigmatism; (**B**) comparison of preoperative and postoperative refractive astigmatism for 6.0 mm High Astigmatism; (**C**) postoperative refractive astigmatism prediction error for 6.0 mm Low Astigmatism; (**D**) postoperative refractive astigmatism prediction error for 6.0 mm High Astigmatism.

**Figure 4 jcm-10-03776-f004:**
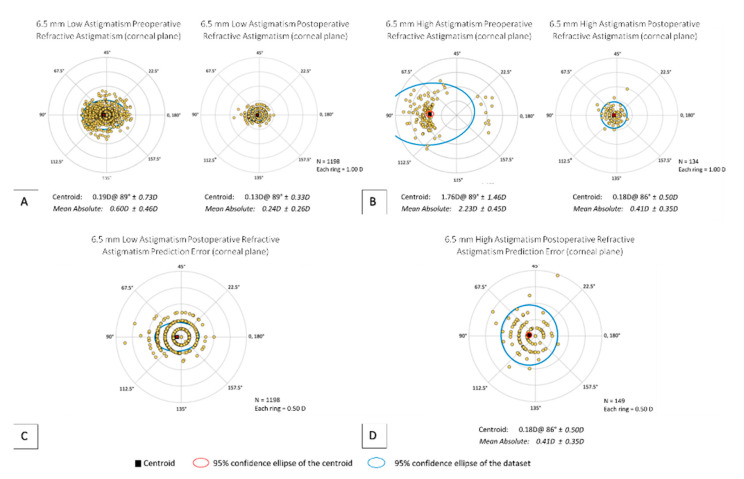
Vector analysis of 6.5 mm Low and High Astigmatism Subgroups. (**A**) Comparison of preoperative and postoperative refractive astigmatism for 6.5 mm Low Astigmatism; (**B**) comparison of preoperative and postoperative refractive astigmatism for 6.5 mm High Astigmatism; (**C**) postoperative refractive astigmatism prediction error for 6.5 mm Low Astigmatism; (**D**) postoperative refractive astigmatism prediction error for 6.5 mm High Astigmatism.

**Figure 5 jcm-10-03776-f005:**
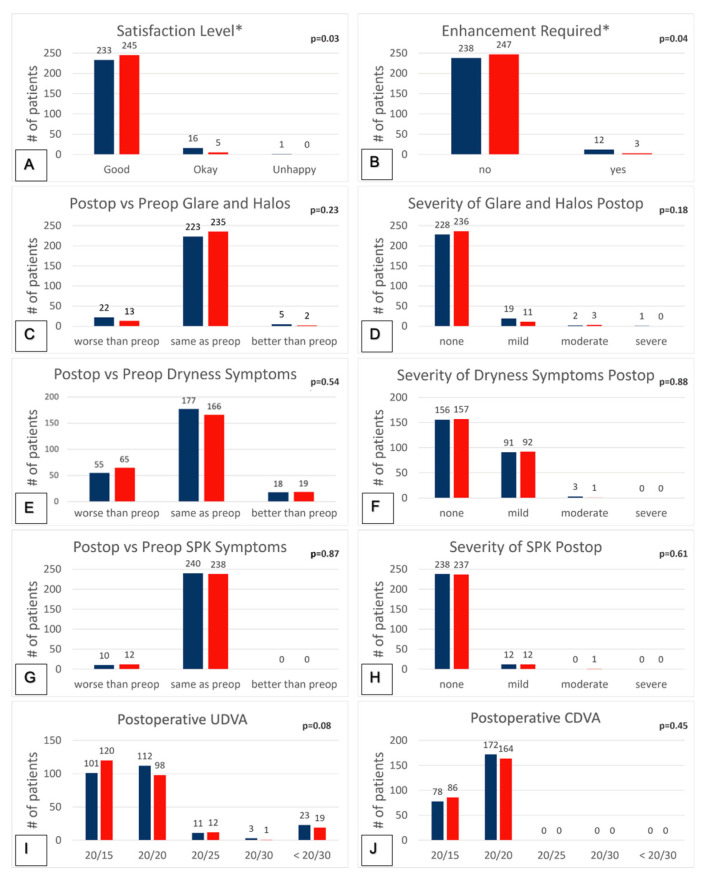
Comparison of patient−reported subjective symptoms between 6.0 and 6.5 mm groups. (**A**) Satisfaction level categorized as “good”, “okay”, and “unhappy”; (**B**) enhancements pursued postoperatively; (**C**) comparison of preoperative and postoperative glare and halos; (**D**) severity of postoperative glare and halos; (**E**) comparison of preoperative and postoperative dry eye symptoms; (**F**) severity of postoperative dry eye symptoms; (**G**) comparison of preoperative and postoperative superficial punctate keratitis (SPK); (**H**) severity of postoperative SPK; (**I**) postoperative UCVA characterized by Snellen; (**J**) postoperative CDVA characterized by Snellen. * statistically significant *p* < 0.05 using a Pearson’s Chi-square or Fisher’s exact test with post-hoc Hochberg method.

**Table 1 jcm-10-03776-t001:** Preoperative baseline characteristics.

Demographics	6.0 mm Mean ± SD|% (*n*)	6.5 mm Mean ± SD|% (*n*)	*p*-Value
Patients/Eyes (*n*)	1332	1332	
Age (years)	34.9 ± 8.3 (18–66)	34.9 ± 8.5 (18–64)	0.80
Male/Female	48.8% (650)/51.2% (682)	52% (693)/48% (639)	0.10
Monovision/Full Distance	7.5% (100)/92.5% (1232)	7.4% (99)/92.6% (1233)	0.10
Parameter	Mean ± SD (range)	Mean ± SD (range)	*p*-value
MRSE (D)	−3.84 ± 1.90 (−0 to −0.5)	−3.84 ± 1.89 (−9 to −0.625)	0.98
Manifest Sphere (D)	−3.41 ± 1.93 (−8.5 to 0)	−3.42 ± 1.92 (−8.5 to 0)	0.94
Manifest Cylinder (D)	−0.85 ± 0.75 (−4.25 to 0)	−0.83 ± 0.74 (−4.25 to 0)	0.60
Axis	81.55 ± 64.37 (0 to 180)	80.51 ± 68.17 (0 to 180)	0.68

Abbreviations: D (diopters); CDVA (best corrected visual acuity); MRSE (manifest refractive spherical equivalent). Statistically significant if *p* < 0.05 using a two-tailed independent samples *t*-test.

**Table 2 jcm-10-03776-t002:** Postoperative results.

Parameter	6.0 mm (*n* = 1332 Eyes) Mean ± SD (Range)	6.5 mm (*n* = 1332 Eyes) Mean ± SD (Range)	*p*-Value
UDVA	−0.02 ± 0.14 (−0.3 to 1.1)	−0.02 ± 0.13 (−0.3 to 1)	0.42
CDVA	−0.06 ± 0.06 (−0.3 to 0.18)	−0.06 ± 0.06 (−0.12 to 0.18)	0.78
MRSE (D)	−0.16 ± 0.37 (−2.5 to 1.25)	−0.12 ± 0.35 (−2.13 to 1.38)	0.01 *
Manifest Sphere (D)	−0.03 ± 0.36 (−2 to 1.5)	−0.01 ± 0.35 (−1.75 to 1.5)	0.01 *
|∆ SEQ| (D)	0.22 ± 0.23 (0 to 2)	0.19 ± 0.20 (0 to 1.38)	<0.001 *
Manifest Cylinder (D)	−0.26 ± 0.28 (−1.75 to 0.75)	−0.26 ± 0.28 (−2 to 0)	0.70
Axis	54.43 ± 66.46 (0 to 180)	55.21 ± 69.78 (0 to 180)	0.77
|Angle of error| (°)	9.03 ± 16.1 (0 to 90)	8.56 ± 14.93 (0 to 90)	0.47

Abbreviations: D (diopters); UDVA (uncorrected distance visual acuity); CDVA (best corrected visual acuity); MRSE (manifest refractive spherical equivalent); |∆ SEQ| (change in mean spherical equivalent); * statistically significant *p* < 0.05 using a two-tailed independent samples *t*-test.

**Table 3 jcm-10-03776-t003:** Postoperative results stratified by severity of myopia.

Parameter	Low Myopia (≥0 D to <−3.0 D)			Moderate Myopia (≥−3.0 D to <−6.0 D)			High Myopia (≥−6.0 D to <−9.0 D)		
	6.0 mm (*n* = 588 Eyes) Mean ± SD (Range)	6.5 mm (*n* = 588 Eyes) Mean ± SD (Range)	*p*-Value	6.0 mm (*n* = 557 Eyes) Mean ± SD (Range)	6.5 mm (*n* = 557 Eyes) Mean ± SD (Range)	*p*-Value	6.0 mm (*n* = 187 Eyes) Mean ± SD (Range)	6.5 mm (*n* = 187 Eyes) Mean ± SD (Range)	*p*−Value
UDVA	−0.02 ± 0.14 (−0.12 to 0.78)	−0.02 ± 0.14 (−0.3 to 1)	0.86	−0.01 ± 0.1 (−0.12 to 1.1)	−0.02± 0.13 (−0.13 to 0.9)	0.26	−0.02 ± 0.12 (−0.3 to 0.9)	−0.02± 0.09 (−0.12 to 0.4)	0.34
CDVA	−0.07 ± 0.06 (−0.12 to 0.1)	−0.06 ± 0.06 (−0.12 to 0.1)	0.12	−0.06 ± 0.06 (−0.12 to 0.18)	−0.07 ± 0.06 (−0.12 to 0.1)	0.14	−0.05 ± 0.07 (−0.13 to 0.18)	−0.06 ± 0.07 (−0.12 to 0.18)	0.20
MRSE (D)	−0.17 ± 0.35 (−2 to 0.875)	−0.14 ± 0.33 (−1.88 to 0.75)	0.21	−0.18 ± 0.39 (−2.5 to 1.25)	−0.14 ± 0.33 (−1.88 to 0.75)	0.03 *	−0.11 ± 0.3 (−1.63 to 0.88)	−0.05 ± 0.37 (−1.88 to 1.25)	0.18
Manifest Sphere (D)	−0.05 ± 0.33 (−2 to 1)	−0.03 ± 0.33 (−1.75 to 0.75)	0.94	−0.04 ± 0.38 (−2 to 1.5)	−0.03 ± 0.33 (−1.75 to 0.75)	0.02 *	−0.04 ± 0.38 (−2 to 1.5)	−0.03 ± 0.33 (−1.75 to 0.75)	0.06
|∆ SEQ| (D)	0.20 ± 0.23 (0 to 2)	0.17 ± 0.18 (0 to 1)	0.01 *	0.23 ± 0.22 (0 to 1.25)	0.18 ± 0.20 (0 to 1.38)	<0.001 *	0.06 ± 0.06 (0 to 0.18)	0.06 ± 0.06 (0 to 0.18)	0.76
Manifest Cylinder (D)	−0.24 ± 0.26 (−1.25 to 0)	−0.22 ± 0.25 (−1.75 to 0)	0.60	−0.28 ± 0.28 (−1.5 to 0)	−0.22 ± 0.25 (−1.75 to 0)	0.92	−0.28 ± 0.28 (−1.5 to 0)	−0.22 ± 0.25 (−1.75 to 0)	0.25
Axis	51.88 ± 64.09 (0 to 180)	50.27 ± 68.11 (0 to 180)	0.68	54.64 ± 66.52 (0 to 180)	50.27 ± 68.11 (0 to 180)	0.61	54.64 ± 66.52 (0 to 180)	50.27 ± 68.11 (0 to 180)	0.57
|Angle of error| (°)	7.27 ± 14.9 (0 to 90)	5.46 ± 10.48 (0 to 90)	0.50	9.41 ± 16.03 (0 to 88.2)	10.42 ± 17.25 (0 to 90)	0.35	13.45 ± 19.17 (0 to 77.8)	13.25 ± 17.63 (0 to 84.7)	0.92

Abbreviations: D (diopters); UDVA (uncorrected distance visual acuity); CDVA (best corrected visual acuity); MRSE (manifest refractive spherical equivalent); |∆ SEQ| (change in mean spherical equivalent); * statistically significant *p* < 0.05 using a two-tailed independent samples *t*-test.

**Table 4 jcm-10-03776-t004:** Postoperative results stratified by severity of astigmatism.

Parameter	Low Cylinder (≥0 D to <−2.0 D)			High Cylinder (≥−2 D to <−4 D)		
	6.0 mm (*n* = 1183 Eyes) Mean ± SD (Range)	6.5 mm (*n* = 1198 Eyes) Mean ± SD (Range)	*p*-Value	6.0 mm (*n* = 148 Eyes) Mean ± SD (Range)	6.5 mm (*n* = 134 Eyes) Mean ± SD (Range)	*p*-Value
UDVA	−0.02± 0.14 (−0.3 to 1.1)	−0.03 ± 0.13 (−0.3 to 1)	0.28	−0.002 ± 0.12 (−0.12 to 0.48)	−0.01 ± 0.14 (−0.12 to 0.55)	0.36
CDVA	−0.06± 0.06 (−0.12 to 0.18)	−0.07 ± 0.06 (−0.12 to 0.18)	0.536	−0.06 ± 0.07 (−0.12 to 0.18)	−0.05 ± 0.06 (−0.12 to 0.18)	0.28
MRSE (D)	−0.16 ± 0.38 (−2.5 to 0.88)	−0.12 ± 0.35 (−2.13 to 1.38)	0.002 *	−0.18 ± 0.35 (−1.38 to 1.25)	−0.20 ± 0.41 (−1.88 to 0.63)	0.68
Manifest Sphere (D)	−0.04 ± 0.36 (−2 to 1)	−0.01 ± 0.34 (−1.75 to 1.5)	0.002 *	−0.03 ± 0.33 (−1.25 to 1.5)	−0.003 ± 0.39 (−1.5 to 0.75)	0.56
|∆ SEQ|	0.21 ± 0.23 (0 to 2)	0.18 ± 0.19 (0 to 1.38)	<0.001 *	0.28 ± 0.26 (0 to 1.38)	0.20± 0.20 (0 to 1)	0.01 *
Manifest Cylinder (D)	−0.24 ± 0.26 (−1.25 to 0.75)	−0.24 ± 0.26 (−1.75 to 0)	0.93	−0.42 ± 0.35 (−1.75 to 0)	−0.41 ± 0.31 (−2 to 0)	0.77
Axis	53.5 ± 66.35 (0 to 180)	53.02 ± 69.05 (0 to 180)	0.86	61.81 ± 65.7 (0 to 179)	74.8 ± 69.6 (0 to 180)	0.11
|Angle of error| (°)	9.89 ± 17.10 (0 to 90)	9.22 ± 15.71 (0 to 90)	0.36	0.04 ± 4.81 (−18.1 to 19.19)	0.45 ± 13.42 (−13.42 to 20.71)	0.50

Abbreviations: D (diopters); UDVA (uncorrected distance visual acuity); CDVA (best corrected visual acuity); MRSE (manifest refractive spherical equivalent); |∆ SEQ| (change in mean spherical equivalent); * statistically significant *p* < 0.05 using a two-tailed independent samples *t*-test.

## Data Availability

The data presented in this study are available on request from the corresponding author. The data are not publicly available due to concern for maintaining patient privacy.

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
