# Peer review of "Comparison of 6.0 mm versus 6.5 mm Optical Zone on Visual Outcomes after LASIK"

_jcm, 2021, doi:10.3390/jcm10173776_

Round 1
Reviewer 1 Report
This is an excellent article and I can only congratulate the authors and try to provide a couple of suggestions. This study presents a large number of patients, with an homogeneous groups and a very nice methodology.
The authors demonstrate that 2 different optical zones (6.0 and 6.5 mm) provide efficacy and safety with the WL EX500. As they themselves reflect while some statistically significant differences exist between the two groups, especially with respect to the spherical equivalent, the outcomes with both optical zone sizes indicate that most patients achieve good postoperative results.
As minor changes i would suggest:
-Being a retrospective study, I would suggest that the authors mention that a prospective study and even better an RCT would be better to corroborate some biases that have been found here, even understanding the difficulty of being able to carry it out.
-Finally, in the section '' Patient − Reported Outcomes and Qualitative Data Collection '' it is not clear to me why they use 250 patients per group instead of 1132, I imagine it is due to its retrospective nature. I suggest they try to explain a little more. I would also add at what point in time the patients were contacted to ask about these symptoms (glare, halos etc) also the SPK. I would also add this fact to the limitations, since although its results are very interesting, they could be slightly biased
Reviewer 2 Report
The manuscript is overall well written and the study well reported. The major concerns regard the lack of significant parameters such as the contrast sensitivity and the spherical aberration. The latter, in particular, is very correlated with the optical zone and a potential game changer in refrative surgery. The authors need to explain the surgical criteria for using a 6.0 or 6.5 mm optical zone. Line 328: 1323 patients is actually 1332 patients.Author Response
Please see the attachment.
